# Long-Term Results of Posterior Vertebral Column Resection for Severe Thoracolumbar Kyphosis with Achondroplastic Patients: A Case Series

**DOI:** 10.3390/medicina58050605

**Published:** 2022-04-27

**Authors:** Masato Tanaka, Tsang-Tung Chan, Haruo Misawa, Koji Uotani, Shinaya Arataki, Tomoyuki Takigawa, Tetsuro Mazaki, Yoshihisa Sugimoto

**Affiliations:** 1Department of Orthopaedic Surgery, Okayama University Hospital, Okayama 700-0914, Japan; ctt405@gmail.com (T.-T.C.); misharuo@gmail.com (H.M.); coji.uo@gmail.com (K.U.); araoyc@gmail.com (S.A.); takigawa2004@yahoo.co.jp (T.T.); umaumauman@hotmail.co.jp (T.M.); sugi.ort@gmail.com (Y.S.); 2Department of Orthopaedic Surgery, Okayama Rosai Hospital, Okayama 702-8055, Japan; 3Department of Orthopaedics and Traumatology, Tseung Kwan O Hospital, Hong Kong; 4Department of Orthopaedic Surgery, Kobe red Cross Hospital, Hyogo 651-0073, Japan; 5Department of Orthopaedic Surgery, Okayama Municipal Hospital, Okayama 700-8557, Japan; 6Department of Orthopaedic Surgery, Kawasaki Medical University Hospital, Okayama 701-0192, Japan

**Keywords:** achondroplasia, vertebral column osteotomy, long-term follow-up, navigation

## Abstract

*Background and Objectives*: Thoracolumbar kyphosis is one of the most frequent skeletal manifestations in patients with achondroplasia. Few papers have been published on the surgical treatment of this condition, especially in skeletally mature patients. With this study, we presented a retrospective case series of long-term surgical results for achondroplastic patients with severe thoracolumbar kyphosis. This study was conducted to evaluate the outcome of surgical treatment for thoracolumbar kyphosis in patients associated with achondroplasia presenting with paraparesis. *Materials and Methods*: Three patients with achondroplasia who developed neurologic deficits due to severe thoracolumbar kyphosis and underwent surgical treatment were evaluated (mean age 22.3 years; mean follow-up 9.3 years). All patients were treated with posterior vertebral column resection (p-VCR) of hypoplastic apical vertebrae with a cage and segmental instrumentation. Neurologic outcomes (JOA scores), correction of kyphosis, and operative complications were assessed. *Results*: All patients had back pain, neurological deficits, and urinary disturbance before surgery. The average preoperative JOA score was 8.3/11 points, which was improved to 10.7/11 points at the final follow-up (mean recovery rate 83%). All patients obtained neurologic improvement after surgery. The mean preoperative kyphotic angle was 117° (range 103°–126°). The postoperative angles averaged 37° (range 14°–57°), resulting in a mean correction rate of 67%. All patients had postoperative complications such as rod breakage and/or surgical site infection. *Conclusions*: The long-term results of p-VCR were acceptable for treating thoracolumbar kyphosis in patients with achondroplasia. To perform this p-VCR safely, spinal navigation and neuromonitoring are inevitable when resecting non anatomical fused vertebrae and ensuring correct pedicle screw insertion. However, surgical complications such as rod breakage and surgical site infection may occur at a high rate, making informed consent very important when surgery is indicated.

## 1. Introduction

Achondroplasia is the most common skeletal dysplasia with short limbs caused by a mutation in the fibroblast growth factor receptor-3 gene [1] with incidences as high as 1 per 15,000 births [2]. The pedicles are short, particularly in the thoracolumbar region, and there is narrowing of the interpedicular spaces of the lumbar vertebrae [3]. Patients with achondroplasia occasionally require surgeries such as foramen magnum decompression [4] or lumbar canal decompression [5] due to neural canal stenosis. Furthermore, diastrophic dwarfism is associated with high rates of scoliosis and kyphosis [6].

Achondroplasia patients with severe progressive neurologic symptoms and spinal deformity may mandate surgical intervention [6]. Pedicle subtraction osteotomy (PSO) and vertebral column resection (VCR) are the most powerful procedures to correct this kind of spinal sagittal malalignment [7,8]. Few reports describe the positive results of correction osteotomy for achondroplasia patients [9,10]. Furthermore, there are no long-term follow-up results of osteotomy for these patients. Currently, navigational spinal surgery for alignment correction has been receiving increasing attention as a method of procedure that is minimally invasive and lowers radiation levels [11,12]. We hereby describe the more than 9-year postoperative follow-up results of three cases of achondroplasia patients who underwent VCR performed utilizing spinal navigation.

## 2. Materials and Methods

This study was approved by the ethics committee of our institute (No. 340). Necessary consents were obtained from the patients. All surgeries were performed at Okayama University Hospital between October 2009 and January 2011. Three patients with achondroplasia who developed neurologic deficit due to severe thoracolumbar kyphosis and underwent surgical treatment were evaluated retrospectively (mean age 22.3 years; mean follow-up 9.3 years). These patients were treated with posterior vertebral column resection (p-VCR) of hypoplastic apical vertebrae with a cage and segmental instrumentation. Neurological outcomes (JOA scores), correction of kyphosis, and operative complications were assessed.

## 3. Results

All patients had back pain, neurological deficits, and urinary disturbance before surgery. The average preoperative JOA score was 8.3/11 points, which was improved to 10.7/11 points at the final follow-up (mean recovery rate 83%). All patients obtained neurologic improvement after surgery. The mean preoperative kyphotic angle was 117° (range 103°–126°). The postoperative angles averaged 37° (range 14°–57°), a mean correction rate of 67%. All patients had postoperative complications, such as rod breakage and/or surgical site infection (Table 1).

### 3.1. Case 1—16-Year-Old Girl, Achondroplasia, Thoracolumbar Kyphosis 126°

#### 3.1.1. Patient History

A 7-year-old girl with achondroplasia was referred to our orthopedic department with low back pain, bilateral leg numbness, and urinary disturbance. She was followed up at our department for 9 years due to thoracolumbar kyphosis. The patient had a urinary disturbance shortly before the operation.

#### 3.1.2. Physical Examination

She was diagnosed with short-limbed dwarfism and her height was 118 cm (7.6 SD). She had short stubby trident hands and a concave facial profile was also noticed. Hyporeflexia of bilateral legs was detected, Babinski reflex was positive on both sides, and the patient could not sleep in the supine position due to severe thoracolumbar kyphosis. The patient had slight muscle weakness in her bilateral legs. The result for the manual muscle test (MMT) of her bilateral legs was 4 for both legs. (Figure 1A–C).

#### 3.1.3. Preoperative Imaging

Radiograms at the initial visit demonstrated there was no scoliosis, but there was severe kyphotic deformity of the thoracolumbar area; T11–L2 local kyphotic angle was 76°, sagittal vertical axis (SVA) 55 mm, pelvic tilt (PT) 32°, and PI 52° (Figure 1D,E). Preoperative radiograms showed progression of kyphosis to be 126° (Figure 1F,G). Preoperative computer tomography (CT) and magnetic resonance imaging (MRI) revealed wedged vertebra (T12 and L1) and stenosis at the thoracolumbar junction (Figure 2).

#### 3.1.4. Surgery

T12 and L1 posterior vertebral column resection (p-VCR) with a mesh cage (PYRAMESH^®^, Medtronic, Medtronic Sofamor Danek, Minneapolis, MN, USA) replacement was performed under neuromonitoring on this patient (Figure 3). First, pedicle screws (CD Horizon Legacy ^®^ 5.5 mm, Medtronic, Medtronic Sofamor Danek, Minneapolis, MN, USA) for T9, T10, T11, L2, and L3 were inserted under navigational guidance. After putting in a unilateral temporary rod to protect the spinal cord from injury due to unintentional spinal movement, T12 and L1 vertebrae were resected carefully. Afterwards, another rod was applied and T9-L3 posterior corrective fusion via massive iliac bone graft was performed; stability was reinforced by sublaminar tapes and hooks (Figure 4A–D). The surgical time was 9 h and 43 min and the estimated blood loss was 4200 mL. No immediate postoperative complications or neurological compromise was reported.

#### 3.1.5. Postoperative Imaging

The postoperative radiograms showed good sagittal alignment; T11-L2 local kyphotic angle was improved from 126° to 14°and SVA was 31 mm with PT of 16° (Figure 4E,F).

#### 3.1.6. Follow-Up Results

On post operative day 10 there was a deep surgical site infection and the patient underwent debridement surgery. Two weeks later, the infection was healed and the MMT of her bilateral legs became normal. She was nearly capable of full activity at one month. She could lay down in a supine position and her daily activity became normal by her 10-year follow-up. The JOA score became 11/11 from 7/11 (recovery rate 100%). The final follow-up radiograms and CT displayed a good sagittal alignment and solid bony fusion (Figure 5).

### 3.2. Case 2—25-Year-Old Man, Achondroplasia, Thoracolumbar Kyphosis 123°

#### 3.2.1. Patient History

A 25-year-old man with achondroplasia was referred to our orthopedic department with low back pain, bilateral leg pain, and gait disturbance. He underwent leg lengthening surgery at 12 years of age.

#### 3.2.2. Physical Examination

He was diagnosed with short-limbed dwarfism and his height was 133 cm (6.8 SD). He had short stubby trident hands and a concave facial profile was also noticed. He had slight muscle weakness in his right leg (MMT 4). The patient could not sleep in the supine position due to severe thoracolumbar kyphosis (Figure 6 A,B).

#### 3.2.3. Preoperative Imaging

Preoperative radiograms showed severe kyphotic deformity of the thoracolumbar area; T11-L2 local kyphotic angle was 123° (Figure 6C,D). Preoperative radiograms showed progression of kyphosis to be 123°, SVA 62 mm, PT 25°, and PI 27° while preoperative MRI revealed stenosis at the thoracolumbar junction (Figure 6E).

#### 3.2.4. Surgery

T12 and L1 p-VCR with a mesh cage (PYRAMESH^®^, Medtronic, Medtronic Sofamor Danek, Minneapolis, MN, USA) replacement was performed under neuromonitoring on this patient (Figure 7A,B). T9-L3 posterior corrective fusion via massive iliac bone graft with pedicle screws (CD Horizon Solera 6.0 mm, Medtronic, Medtronic Sofamor Danek, Minneapolis, MN, USA) and sublaminar tapes was performed. The surgical time was 8 h and 20 min and the estimated blood loss was 1720 mL. No immediate postoperative complications or neurological compromise was reported.

#### 3.2.5. Postoperative Imaging

The postoperative radiogram showed good sagittal alignment; T11-L2 local kyphotic angle was improved from 123° to 38°, SVA 42 mm, PT 3°, and PI 27° (Figure 7C,D).

#### 3.2.6. Follow-Up Results

Two years later, he underwent revision surgery due to rod breakage. He had slight leg numbness but his daily activity became almost normal by his 9-year follow-up. The JOA score became 10/11 from 8/11 (recovery rate 67%). The final follow-up radiograms and CT revealed a good sagittal alignment and solid bony fusion (Figure 8).

### 3.3. Case 3—26-Year-Old Man, Achondroplasia, Thoracolumbar Kyphosis 103°

#### 3.3.1. Patient History

A 26-year-old man with achondroplasia was referred to our orthopedic department with low back pain, bilateral leg numbness, and pollakiuria. He underwent leg lengthening surgery at 15 years of age.

#### 3.3.2. Physical Examination

He was diagnosed with short-limbed dwarfism and his height was 135 cm (6.6 SD). He had no muscle weakness in his right leg but severe numbness of both feet (Figure 6A,B).

#### 3.3.3. Preoperative Imaging

Preoperative radiograms showed severe kyphotic deformity of the thoracolumbar area; T11–L2 local kyphotic angle was 103°, SVA 60 mm, PT 27°, and PI 81° (Figure 9A-C). Preoperative CT showed T11-L1 wedged deformity (Figure 9D,E) and MRI revealed stenosis at the T11/12 and T12/L1 level (Figure 10).

#### 3.3.4. Surgery

T12 and L1 p-VCR with an expandable cage (T2 Altitude^®^, Medtronic, Medtronic Sofamor Danek, Minneapolis, MN, USA) replacement was performed under neuromonitoring on this patient (Figure 10A). T9-L3 posterior corrective fusion via massive iliac bone graft with pedicle screws (CD Horizon Solera 6.0 mm, Medtronic, Medtronic Sofamor Danek, Minneapolis, MN, USA) and sublaminar tapes was performed. The surgical time was 8 h and 10 min and the estimated blood loss was 3500 mL. No immediate postoperative complications or neurological compromise was reported.

#### 3.3.5. Postoperative Imaging

The postoperative radiogram showed good sagittal alignment; T11-L2 local kyphotic angle was improved from 103° to 57°, SVA 41 mm, PT 27°, and PI 82° (Figure 11B,C). CT revealed good expandable vertebral cage positioning (Figure 11D).

#### 3.3.6. Follow-Up Results

One year and three years later, he underwent two revision surgeries due to rod breakage and surgical site infection. He had no back pain and his daily activity was normal at the 9-year follow-up. The JOA score became 11/11 from 9/11 (recovery rate 100%). The final follow-up radiograms and CT revealed good sagittal alignment and solid bony fusion (Figure 12).

## 4. Discussion

Achondroplasia is the most common cause (more than 90%) of disproportionate short stature [13]. The prevalence is estimated to be between 1 in 15,000 and 1 in 40,000 live births [11,14,15,16]. It is caused by a mutation in the FGFR3 gene which results in the suppression of chondrocyte differentiation and cartilage matrix production and proliferation. The bone formation is impaired due to defective endochondral ossification, and thus, clinically manifests as classical clinical features, such as rhizomelic short stature, macrocephaly, depressed nasal bridge, and trident hands. Furthermore, it causes several characteristic spinal problems that warrant spinal surgeons’ attention. These include craniocervical junction stenosis that could result in early childhood death, a narrowed spinal canal that occurs at all levels of the vertebral column, and thoracolumbar kyphosis (TLK) that is present in nearly all achondroplasia infants. Among them, thoracolumbar kyphosis (TLK) is present in nearly all achondroplasia infants [3,11,13,17,18].

This early infantile thoracolumbar kyphosis is non-congenital and is not associated with any primary structural defect of the vertebrae [16,19]. The management of TLK deformity includes a detailed history as well as a neurological examination and imaging assessment. The kyphotic angle in sitting and supine positions as measured in serial lateral X-rays is an invaluable tool for detecting fixed TLK components and monitoring progression. The whole spine PA and lateral standing X-rays provide information on global sagittal alignment. MRI provides information on the thoracolumbar region as well as common neurological complication areas including the brain and craniocervical junction. Usually, this TLK deformity will progress after the child has adopted sitting at the age of 6–18 months and will spontaneously improve after 1 year of standing and walking [16,19,20]. However, there are around 10–30% of TLK cases that will persist until the age of 5 [11,16,21]. Over time, these persistent TLK deformities may become fixed and are associated with the bullet-shaped wedging vertebral deformity at the apex of the kyphosis [16,20]. Identified risk factors for TLK maintenance or progression include developmental motor delay, radiological parameters of apical vertebral translation, and apical vertebral wedging affecting vertebral height [19,22]. Clinically, symptoms of the TLK deformity include back pain, lower limb weakness, numbness, claudication, and bladder or bowel incontinence. The incidence of these neurological symptoms correlates to the degree of wedging deformity.

A theory has been proposed that unsupported sitting of achondroplasia infants is related to the progression of the TLK deformity. It is believed that during unsupported sitting, the associated abnormal C-shaped sitting posture creates an anomalous gravitational force which causes compression and remodeling of the anterior vertebrae. Based on this theory, measures that aim to prevent the progression of TLK deformity have been developed. These include counselling against unsupported sitting during infancy and the use of TLSO braces in selected cases (>30 degrees fixed component or significant vertebral body deformity). In general, the outcome of these measures would be more favorable if they could be applied at the earlier stages of disease, before the anterior vertebral epiphyseal ring has been disrupted by the anterior compressive force [16,19,22,23,24].

In cases where non-operative management for achondroplasia TLK deformity becomes ineffective, surgical intervention is indicated when there are incapacitating/progressive neurological symptoms or spinal deformity [6,19,24]. The aim of the surgical intervention is to achieve neurological decompression, deformity correction, and prevention of future deformity progression. Various surgical treatment techniques have been described, which include laminectomy for decompression, laminectomy with fusion, pedicle subtraction osteotomy (PSO), and vertebral column resection (VCR). Historically, spinal instrumentation was discouraged because of the risks of neurological complications related to the abnormal vertebral anatomy in achondroplasia patients [3,25]. Currently, with technological advancements in intraoperative imaging, navigation, and intraoperative neuromonitoring, the risk of neurological complications for spinal instrumentation has been reduced [26]. Concerning the choice of implant, pedicle screw fixation is favored due to its ability to achieve higher immediate rigidity and correction of the sagittal plane deformity. If it is necessary to use the sublaminar hook, caution should be taken to avoid causing further narrowing of the spinal canal in the presence of achondroplastic deformities [27,28].

On rare occasions when the TLK deformity is not associated with ventral cord compression, laminectomy alone can be performed for decompression. Owing to the nature of having multilevel spinal stenosis in achondroplasia patients, multilevel laminectomy is often necessary for a good long-term neurological and functional outcome. [29] Instrumented spinal fusion is recommended after multilevel laminectomy to obtain stability and to prevent further progression of kyphosis [30]. In cases where there is ventral compression due to the apex of the TLK deformity, laminectomy alone is not sufficient for treatment. The apex of the TLK deformity can cause injury to the spinal cord due to either direct compression or blood flow disturbance related to compression over the anterior spinal cord vessels. Therefore, circumferential decompression is necessary for adequate decompression over the ventral compression. This can be achieved by performing multilevel laminectomy with deformity correction +/− resection of the apical vertebrae. Among these surgical techniques for TLK deformity correction, PSO and VCR are the two most powerful procedures [7,8]. In PSO, the posterior elements, pedicles, and a triangular wedge of the posterior part of the vertebral body are removed. The sagittal alignment is corrected by shortening the posterior spine with the anterior cortex as a hinge. Although PSO is a powerful technique, there are some limitations. Firstly, because of its anteriorly located correction hinge, the posterior vertebral body needs to be shortened for deformity correction. This shortening may be impossible in some cases with hypoplastic and wedge-shaped apical vertebrae. Furthermore, the shortening may result in kinking of the spinal cord, and thus neurological injury [31]. Secondly, PSO has limited power in coronal plane deformity and rotational deformity.

On the other hand, the VCR provides the most powerful means for 3-D spinal deformity correction in coronal, sagittal, translational, and rotational planes. Compared to PSO, the shortening of the posterior vertebral body is not necessary for deformity correction. In this study, posterior vertebral column resections (p-VCR) were performed on all our achondroplasia patients who were skeletally mature with severe kyphotic deformity of the thoracolumbar region. With instrumentation guided under navigation and intraoperative neuromonitoring in all cases, there was no neurological complication in our series. There was also no dura tear nor CSF leakage, a common complication in spinal surgery performed on achondroplasia patients [32]. The global sagittal alignment was restored in all patients, with the sagittal vertical axis averaging 38 mm (31–41 mm). However, there was one case of early, and one case of late, surgical site infection. All were treated with revision surgery without long-term complications. There were two cases of rod breakage (rate 66%) that occurred at 1 and 2 years after surgery, respectively (Table 1). Revision surgery was performed and all patients in our series eventually achieved solid bony fusion.

Concerning the problem of rod breakage in achondroplasia and TLK cases treated with p-VCR correction, Wang et al. also reported a case series with the rate of rod breakage as high as 43% within 12–18 months postoperatively [33]. Several factors may explain the high rate of rod breakage in the period within 2 years of the VCR procedure. Implants are under high stress due to the extreme instability created by the three-column osteotomy (3CO) of VCR. The rod fracture rate after VCR was reported to be around 10–30% in studies involving non-achondroplasia patients [34,35]. The long fusion construct that is usually required for TLK surgery further increases stress across the 3CO site due to its longer moment arm [36]. Besides the instability problem, the fusion process may also be impaired in achondroplasia patients who have undergone VCR. This may be explained by their defective endochondral ossification process and the absence of direct bone-to-bone surfaces for healing after VCR (unlike PSO). As a result, achondroplasia patients may have a higher risk of nonunion or delayed union which renders the fusion construct susceptible to fatigue failure. In view of the aforementioned conditions, the use of a multi-rod construct may provide better construct stability not only for revision surgery but also for primary procedures. This may potentially lower the risk of rod breakage and improve the fusion time and rate [33,34].

Although surgical complications such as rod breakage and surgical site infection may occur in p-VCR correction surgeries for TLK deformities, all of our patients in this case series achieved good long-term outcomes with solid fusion after appropriate complication management.

There are several limitations of this study. Because of this relatively rare condition, the study number was small. Longer follow-up results and patient-reported outcomes were recommended.

## 5. Conclusions

Posterior vertebral column resection (p-VCR) is one of the most effective surgeries for thoracolumbar kyphosis patients with achondroplasia. To perform this technique safely, spinal navigation and neuromonitoring is inevitable when resecting non-anatomical fused vertebrae and ensuring correct pedicle screw insertion. However, as we are correcting a severe deformity, it is better to opt for multiple rods and adequate bony fusion to avoid complications and revision surgery.

## Figures and Tables

**Figure 1 medicina-58-00605-f001:**
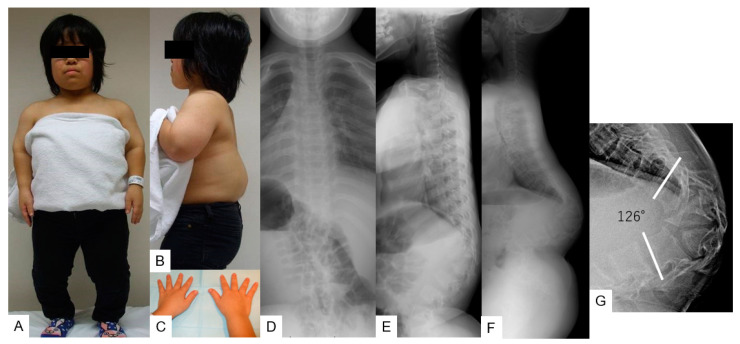
A 16-yearold girl with achondroplasia; preoperative figure and radiograms. (**A**,**B**) The patient’s posture shows dwarfism with short limbs and severe thoracolumbar kyphosis; (**C**) hands are three-pronged (trident) in appearance; (**D**,**E**) radiograms at initial visit; (**F**,**G**) lateral radiogram shows severe sagittal malalignment; T11-L2 local kyphosis 126°, SVA 55 mm, PT 32°, and PI 52°.

**Figure 2 medicina-58-00605-f002:**
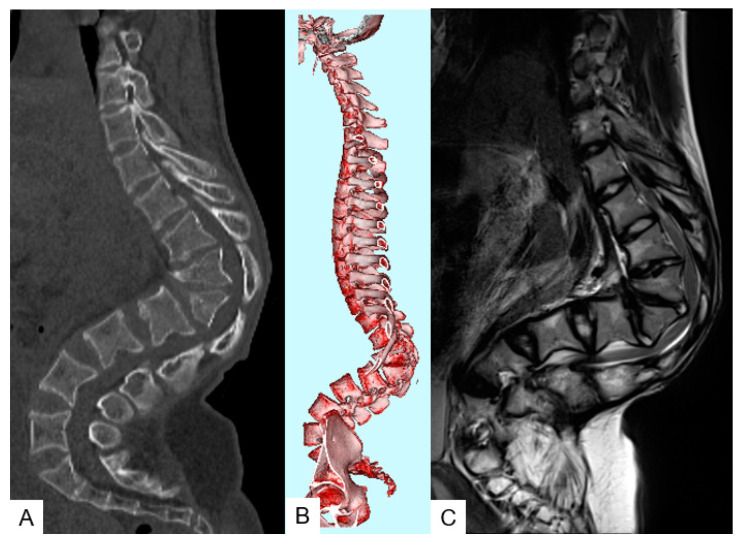
A 16-year-old girl with achondroplasia in preoperative state; preoperative CT and MRI. (**A**) Sagittal reconstruction CT image; (**B**) 3-D CT; (**C**) thoracolumbar sagittal T2-weighted MR imaging shows severe kyphosis and stenosis.

**Figure 3 medicina-58-00605-f003:**
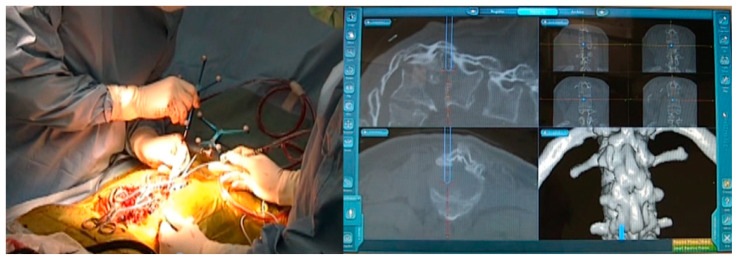
A 16-year-old girl with achondroplasia; intraoperative image and navigation.

**Figure 4 medicina-58-00605-f004:**
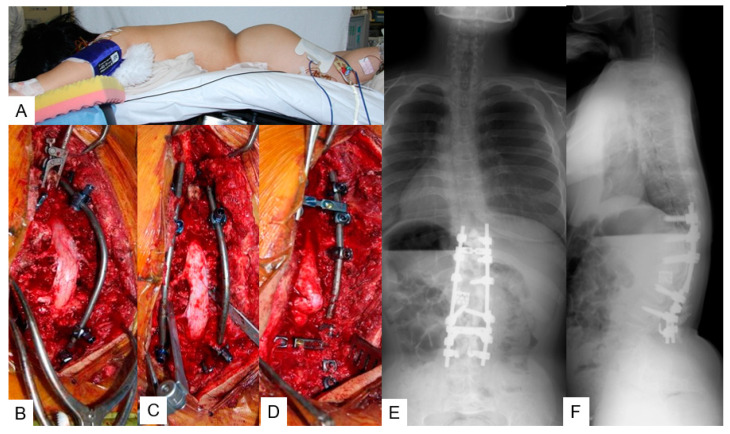
A 16-year-old girl with achondroplasia; intra- and postoperative radiograms. (**A**) Flexible surgical table; (**B**–**D**) intraoperative images; (**E**,**F**) postoperative radiograms.

**Figure 5 medicina-58-00605-f005:**
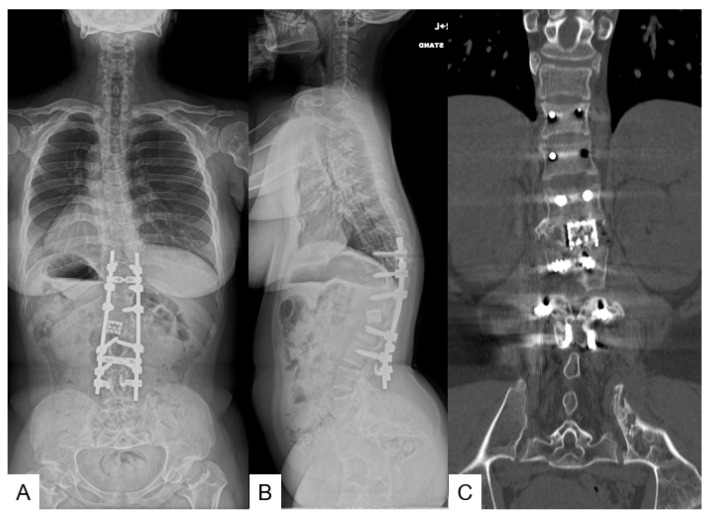
A 16-year-old girl with achondroplasia; final follow-up images. (**A**) Posteroanterior radiogram; (**B**) lateral radiogram; (**C**) coronal reconstruction CT. T11-L2 local kyphosis 14°; SVA 31 mm; PT 16°; PI 52°.

**Figure 6 medicina-58-00605-f006:**
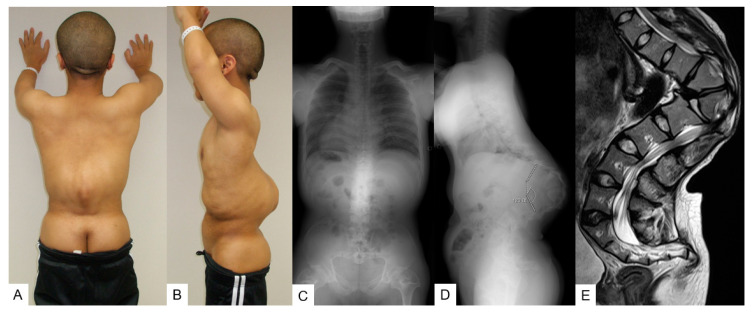
A 25-year-old man with achondroplasia; preoperative figure and radiograms. (**A**,**B**) The patient’s posture; (**C**,**D**) preoperative radiograms show T11-L2 local 123° kyphosis, SVA 62 mm, PT 25°, and PI 27°; (**E**) thoracolumbar sagittal T2-weighted MR imaging shows severe kyphosis and stenosis.

**Figure 7 medicina-58-00605-f007:**
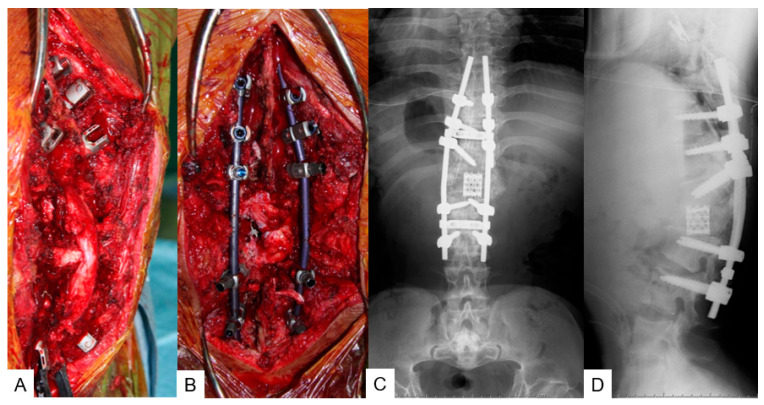
**A** 25-year-old man with achondroplasia; intra- and postoperative radiograms. (**A**,**B**) Intraoperative images; (**C**,**D**) postoperative radiograms.

**Figure 8 medicina-58-00605-f008:**
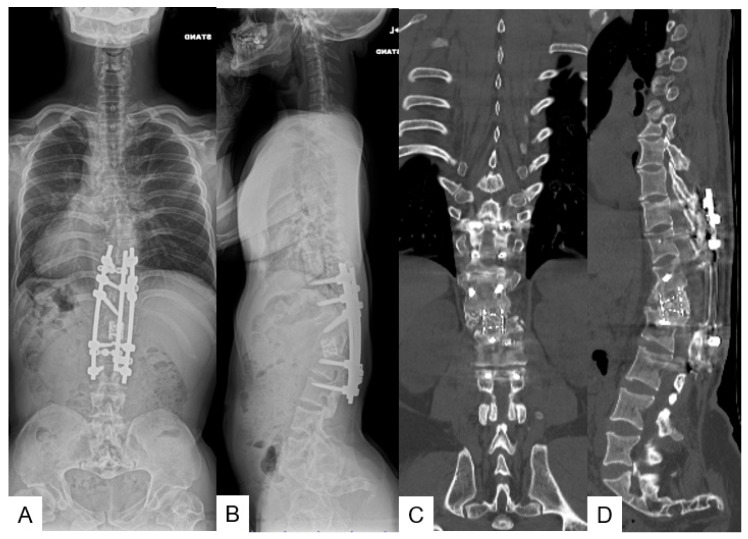
A 25-year-old man with achondroplasia; final follow-up radiograms and CT. (**A**)Anteroposterior radiogram shows good coronal alignment with triple rod-fixation; (**B**) Lateral radiogram indicates acceptable local kyphosis (T11-L2 local 39° kyphosis, SVA 43 mm, PT 5°, PI 27°); (**C**,**D**) Final coronal and sagittal reconstruction CT shows solid bony fusion.

**Figure 9 medicina-58-00605-f009:**
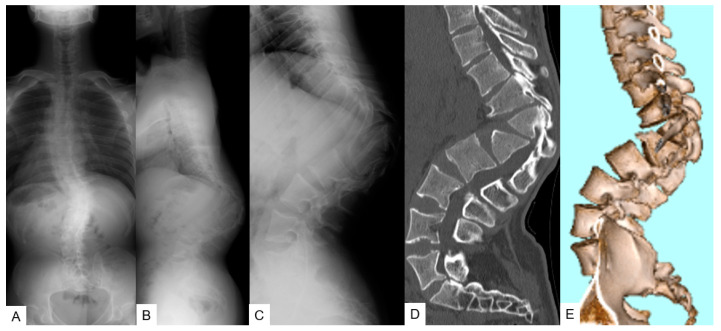
A 26-year-old man with achondroplasia; preoperative radiograms and CT. (**A**) Posteroanterior radiogram; (**B**,**C**) lateral radiogram (T11-L2 local kyphosis 103°, SVA 60 mm, PT 27°, PI 81°); (**D**) sagittal reconstruction CT; (**E**) 3-D CT.

**Figure 10 medicina-58-00605-f010:**
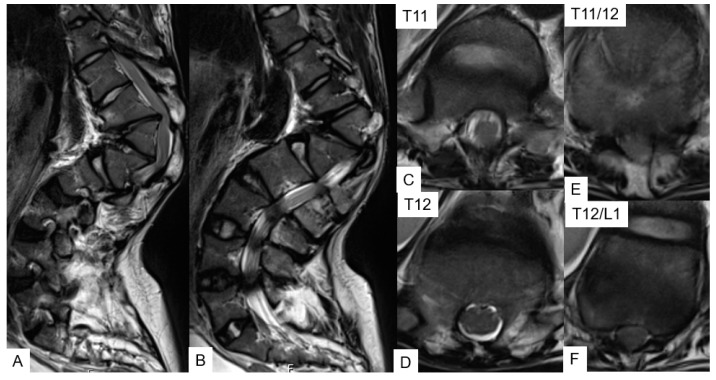
A 26-year-old man with achondroplasia; preoperative MRI. (**A**,**B**) Thoracolumbar sagittal T2-weighted MR imaging; (**C**–**F**) axial T2-weighted MR images show severe stenosis at the T11/12 and L1/2 level.

**Figure 11 medicina-58-00605-f011:**
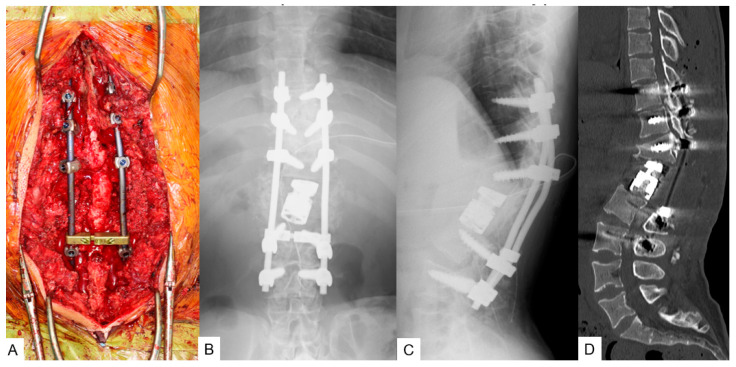
A 26-yearold man with achondroplasia; intra- and postoperative images. (**A**) Intraoperative images; (**B**,**C**) postoperative radiograms; (**D**) sagittal reconstruction CT.

**Figure 12 medicina-58-00605-f012:**
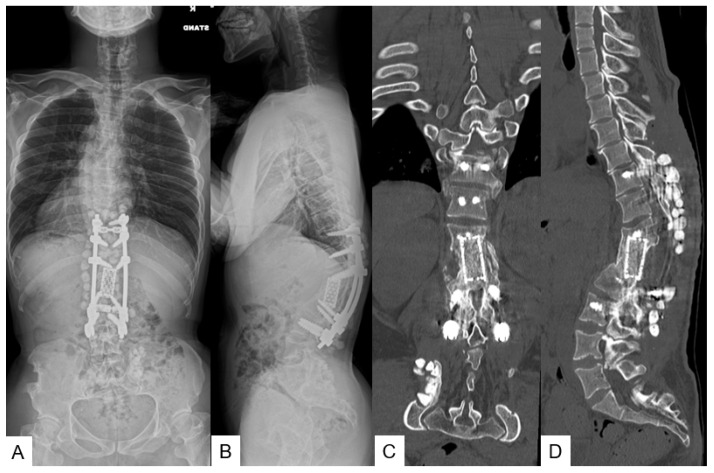
A 26-year-old man with achondroplasia; final follow-up images. (**A**,**B**) Follow-up radiograms show T11-L2 local 58° kyphosis, SVA 32 mm, PT 23°, and PI 82°; (**C**,**D**) follow-up reconstruction CT shows solid bony fusion.

**Table 1 medicina-58-00605-t001:** Summary of clinical results.

No.	JOA Scores Pre-Op	JOA Scores F/U	Years Of FU Post Op	Kyphosis Cobb Pre-Op	Kyphosis Cobb Post-Op	Correction Rate	Fusion Level	p-VCR	Complications
1	7	11	10	126	14	89%	T9-L3	T12,L1	SSI (day 10)
2	8	10	9	123	38	69%	T9-L3	T12,L1	Rod breakage (2 years)
3	9	11	9	103	57	47%	T9-L3	T12, L1	Rod breakage (1 year) SSI (3 years)

SSI: Surgical site infection.

## Data Availability

The data presented in this study are available in the article.

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
