# Peer review of "Long-Term Results of Posterior Vertebral Column Resection for Severe Thoracolumbar Kyphosis with Achondroplastic Patients: A Case Series"

_medicina, 2022, doi:10.3390/medicina58050605_

Round 1
Reviewer 1 Report
Achondroplasia with severe thoracolumbar kyphosis is a rare but serious problem. The authors did a good job treating these 3 cases with p-VCR. This paper is well-written and informative for surgeons who are dealing with this problem.
Although the case number is small (only 3 cases), I still highly recommend accepting and publishing this paper for a rather long-term follow-up and successful treatment outcomes.
Two of three patients in this article received revision surgery for rod breakage. Authors may describe the reasons for rod breakage in these 2 patients. Was rod breakage a result of pseudarthrosis at the p-VCR site or fatigue failure of rods due to movements of intervertebral discs.
Authors may further describe how they did in the revision surgery to achieve solid fusion. Did they use multiple rods or special implants to facilitate fusion at the p-VCR sites?
Author Response
Reviewer 1
Achondroplasia with severe thoracolumbar kyphosis is a rare but serious problem. The authors did a good job treating these 3 cases with p-VCR. This paper is well-written and informative for surgeons who are dealing with this problem.
Although the case number is small (only 3 cases), I still highly recommend accepting and publishing this paper for a rather long-term follow-up and successful treatment outcomes.
Two of three patients in this article received revision surgery for rod breakage. Authors may describe the reasons for rod breakage in these 2 patients. Was rod breakage a result of pseudarthrosis at the p-VCR site or fatigue failure of rods due to movements of intervertebral discs.
We appreciate your important question.
Yes, rod broken site of two cases was at the p-VCR site. We should have applied multiple rods but unfortunately, we didn’t have such a good idea at that time as we have never encountered such complications before hand.
Authors may further describe how they did in the revision surgery to achieve solid fusion. Did they use multiple rods or special implants to facilitate fusion at the p-VCR sites?
Thank you for your valuable question.
As I mentioned above, we did revision surgery with multiple rods and massive bone graft. tocreate proper bony fusion.
Reviewer 2 Report
I commend the authors for their research entitled "Long-term Results of Posterior Vertebral Column Resection for Severe Thoracolumbar Kyphosis with Achondroplastic Patients: A Case Series." The manuscript is interesting, well structured, nicely illustrated and easy to follow. I would suggest the following:
- Materials and Methods: Please provide details of all implants (cages and segmental instrumentation) used at surgeries (material, commercial name, producer, city and state).
- Figures: Figure legends should be written correctly at all figures.
- Case 2: Check Figure 8. It looks like A and B are showing postoperative X-rays with double rods on one side, C and D are showing CT images and there is no E with CT-3D. Also check Figure 9 labels (wrong/no comments for C,D, and E).
- Discussion: State the limitations of your research (e.g. small group of patients). You have used multirod construct at revision in one patient - is this correct? Would you recommend to use such construct in theses cases routinely at all revisions? What about primary procedures?
- Conclusions: Conclusion should be revised and enriched once to reflect the significance of the topic.
Author Response
Reviewer 2
I commend the authors for their research entitled "Long-term Results of Posterior Vertebral Column Resection for Severe Thoracolumbar Kyphosis with Achondroplastic Patients: A Case Series." The manuscript is interesting, well structured, nicely illustrated and easy to follow. I would suggest the following:
Materials and Methods: Please provide details of all implants (cages and segmental instrumentation) used at surgeries (material, commercial name, producer, city and state).
Thank you for your important comments.
These products are all Medtronic products.
We added the name of implants and those detailed as follows;
Case 1 PYRAMESH ® cage, CD Horizon Legacy ® 5.5mm (Medtronic, Medtronic Sofamor Danek, Minneapolis, MN, USA)
Case 2 PYRAMESH ® cage, CD Horizon Solera 6.0mm
Case 3 T2 Altitude ® cage, CD Horizon Solera ® 6.0mm
Figures: Figure legends should be written correctly at all figures.
We appreciate your valuable comment.
We added the sentences according to your advice.
Case 2: Check Figure 8. It looks like A and B are showing postoperative X-rays with double rods on one side, C and D are showing CT images and there is no E with CT-3D. Also check Figure 9 labels (wrong/no comments for C,D, and E).
Thank you for your comments.
We changed the sentences accordingly.
Discussion: State the limitations of your research (e.g. small group of patients). You have used multirod construct at revision in one patient - is this correct? Would you recommend to use such construct in theses cases routinely at all revisions? What about primary procedures?
We appreciate your important comment.
We added the sentences as follow in the discussion part.
We recommend to use multiple rods application and proper bony grafting to achieve solid bony fusion to avoid further complications not only for revision surgery but also for primary procedure.
There are several limitations of this study. Because of this relatively rare condition, the study number was small. Longer follow-up results and patient-reported outcomes were recommended.
Conclusions: Conclusion should be revised and enriched once to reflect the significance of the topic.
Thank you for your thoughtful suggestion
We changed the conclusions as follows;
Posterior vertebral column resection (p-VCR) is one of the most effective surgeries for the thoracolumbar kyphosis patients with achondroplasia. To perform this technique safely, spinal navigation and neuromonitoring is inevitable to resect non anatomical fused vertebrae and correct pedicle screw insertion. However, as we are correcting a severe deformity, it’s better to opt for multiple rods and adequate bony fusion to avoid complications and revision surgery.